# LayoutDIT: Layout-Aware End-to-End Document Image Translation with Multi-Step Conductive Decoder

**Zhiyang Zhang[1,2], Yaping Zhang[1,2], Yupu Liang[1,2], Lu Xiang[1,2]***,
**Yang Zhao[1,2], Yu Zhou[1,3], Chengqing Zong[1,2]**

[1] State Key Laboratory of Multimodal Artificial Intelligence Systems (MAIS),
Institute of Automation, Chinese Academy of Sciences, Beijing, China
[2] School of Artificial Intelligence, University of Chinese Academy of Sciences, Beijing, China
[3] Fanyu AI Laboratory, Zhongke Fanyu Technology Co., Ltd, Beijing, China

zhangzhiyang2020@ia.ac.cn, {yaping.zhang, lu.xiang, yang.zhao, yzhou, cqzong}@nlpr.ia.ac.cn

## Abstract

Document image translation (DIT) aims to translate text embedded in images from one language to another. It is a challenging task that needs to understand visual layout with text semantics simultaneously. However, existing methods struggle to capture the crucial visual layout in real-world complex document images. In this work, we make the first attempt to incorporate layout knowledge into DIT in an end-to-end way. Specifically, we propose a novel **Layout**-aware end-to-end **D**ocument **I**mage **T**ranslation (**LayoutDIT**) with multi-step conductive decoder. A layout-aware encoder is first introduced to model visual layout relations with raw OCR results. Then a novel multi-step conductive decoder is unified with hidden states conduction across three step-decoders to achieve the document translation step by step. Benefiting from the layout-aware end-to-end joint training, our LayoutDIT outperforms state-of-the-art methods with better parameter efficiency. Besides, we create a new multi-domain document image translation dataset to validate the model's generalization. Extensive experiments show that LayoutDIT has a good generalization in diverse and complex layout scenes.

## 1 Introduction

Document image translation (DIT) aims to translate text embedded in scanned documents from one language to another. Automated DIT is a crucial research area for business and academic values. Different from plain text translation, texts on document images are arranged with certain layouts. It requires the model to understand the visual layouts and text semantics simultaneously. Figure 1 showcases visual layout plays a vital role in DIT. Missing or error layout information could result in subsequent translation failure.

---
*Corresponding author.

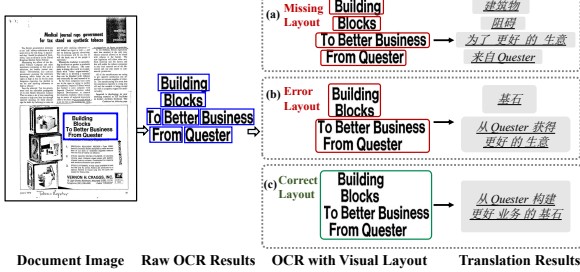

Figure 1: **The role of layout in DIT.** Words on the document image are arranged with certain visual layouts critical for translation. Either a missing or error layout would negatively affect the subsequent translation.

However, existing works struggle to capture the visual layout with a logical reading order on document images (Sable et al., 2023; Long et al., 2022; Hinami et al., 2021; Afli and Way, 2016; Chen et al., 2015; Du et al., 2011). As shown in Figure 2, given a sequence of word pieces extracted by Optical Character Recognition (OCR), existing cascade methods (Hinami et al., 2021) get the rational translation through four separate processes:(1) layout parsing, (2) logical order detection, (3) sentence segmentation, and (4) translation. They are handled as isolated individuals and optimized independently, which leads to severe error propagation. Another problem is that it ignores the semantic correlation shared in different modules.

To address the above problems, in this work, we propose a unified end-to-end trainable **Layout**-aware **D**ocument **I**mage **T**ranslation (**LayoutDIT**) network to map a document image processed by OCR to its document translation. LayoutDIT could simultaneously achieve layout understanding, reading order detection, sentence segmentation, and translation, and share computation and semantic information among these complementary tasks. Specifically, LayoutDIT comprises a layout-aware encoding phase and a multi-step conductive decoding phase. The layout-aware encoding phase aims

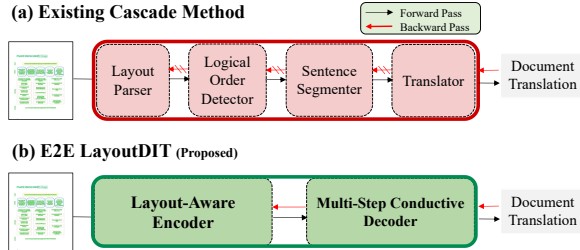

Figure 2: **DIT methods overview.** Given a document image with words and word bounding boxes extracted by OCR, DIT aims to generate the document-level translation in the target language. Different from the traditional cascade framework, LayoutDIT goes beyond with layout-aware end-to-end modeling, showing conceptual conciseness and model compactness.

to *model document image's texts with layouts* by jointly embedding text tokens and their 2D layouts. The multi-step conductive decoding phase is to *capture the reading logic and to get rational translation results*. The decoding phase identifies a chain of three decoding sub-processes: reading order decoding, sentence boundary decoding, and translation decoding. It decodes intermediate inferences for each sub-process to approach the final document translation step by step. Each intermediate decoding step is fulfilled dedicatedly with a step-decoder. All step-decoders are unified into a multi-step decoder with the conduction of hidden states across them, thus enabling LayoutDIT's sub-module interaction and end-to-end training.

We evaluate LayoutDIT's strong ability on the public ReadingBank (Wang et al., 2021) for general-domain document image translation. In addition, we create a new multi-domain Document Image Translation dataset (DITrans), which covers three different document domains to test LayoutDIT's domain-specific capacity. Experimental results show that the proposed LayoutDIT can achieve state-of-the-art translation performance with better parameter efficiency. Further analysis demonstrates LayoutDIT's great power in cross-domain adaptation and few-shot learning, indicating its superiority in these practical DIT scenarios. The contributions of this work are summarized as follows:

- We propose a layout-aware end-to-end document translation framework named LayoutDIT. To the best of our knowledge, this is the first work that incorporates layout information into DIT in an end-to-end way, which shows

conceptual conciseness and better parameter efficiency.
- We devise a novel layout-aware encoder and multi-step conductive decoder to enable layout awareness and sub-module interactions simultaneously, which could share computation among four complementary tasks, including layout parsing, reading order detection, sentence segmentation, and translation.
- A new multi-domain DIT dataset named DITrans is created to evaluate LayoutDIT. Extensive experiments[1] and analysis on public benchmark and our dataset show LayoutDIT's better overall performances and superior capability, especially for low-resource DIT.

## 2   Layout-aware Document Image Translation

In this section, we describe in detail our model architecture, as seen in Figure 3. LayoutDIT consists of a layout-aware encoding phase and a multi-step conductive decoding phase. The former employs a layout-aware encoder for document image understanding by producing its layout-aware representations; the latter generates the document translation following a "divide-and-conquer" philosophy. Concretely, a multi-step conductive decoder is proposed to divide the decoding phase into a chain of three sub-processes and conquers them with dedicated step-decoders, one by one:

- **Decoding Step 1** (Reading order decoding). Given an OCR document image with out-of-order words and word bounding boxes, a re-ordering step-decoder decodes each word's reading order index based on the layout-aware representations.
- **Decoding Step 2** (Sentence boundary decoding). Given an in-order word sequence, a segmentation step-decoder decodes each word's sentence boundary tag to find the begging/end of each sentence.
- **Decoding Step 3** (Translation decoding). A translation decoder decodes the target language translation for each sentence.

The final document translation can be obtained by concatenating all sentences' translations.

---

[1]Code will be released at https://github.com/zhangzhiyang-2020/LayoutDIT.

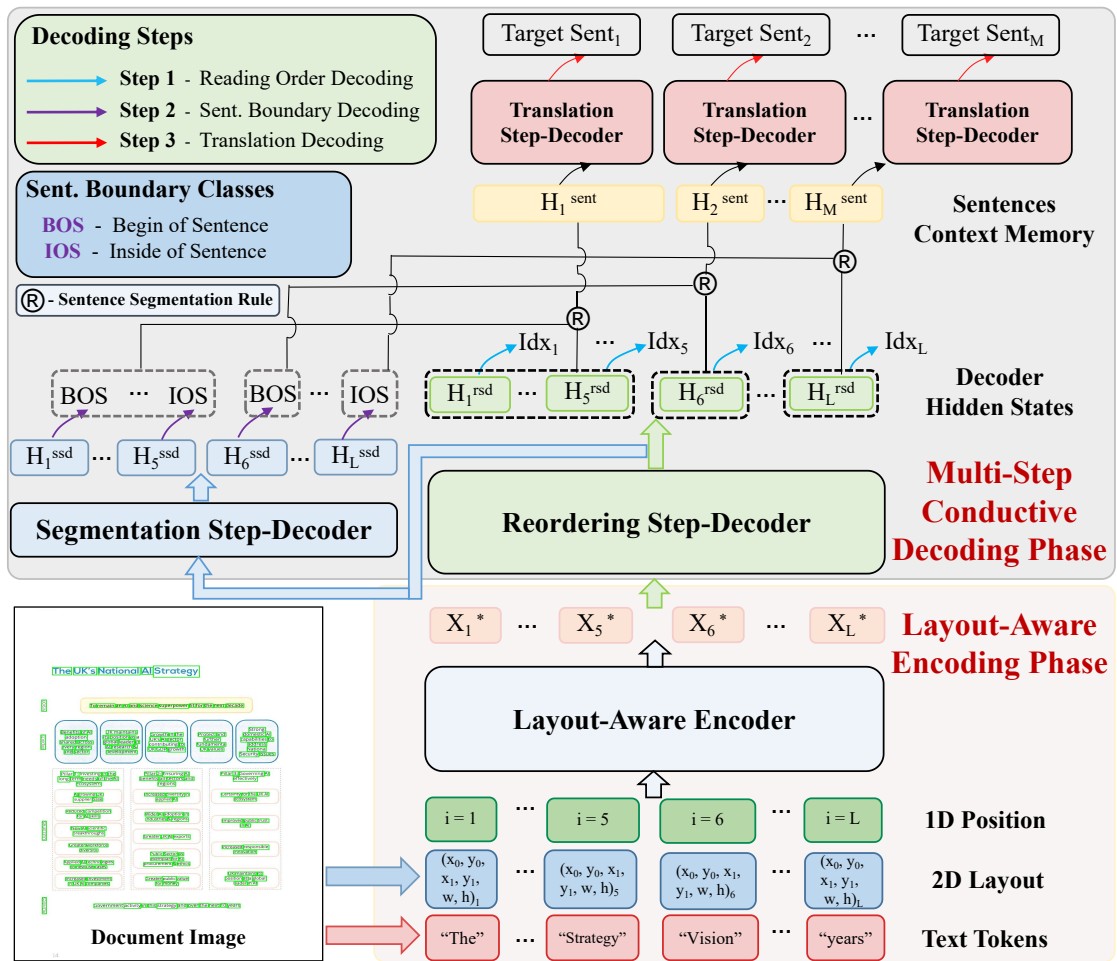

Figure 3: **Proposed DIT framework *LayoutDIT*.** During the layout-aware encoding phase, it jointly encodes a document image's visual layouts and text semantics. During the multi-step conductive decoding phase, it generates the document translation step by step with a chain of decoding sub-processes: reading order decoding, sentence boundary decoding, and translation decoding. Each decoding step is achieved with a function-specific step-decoder. With the conduction of hidden states, all three step-decoders are unified into one multi-step conductive decoder, enabling LayoutDIT's sub-module interaction and end-to-end training.

## 2.1 Document Image and Recognized Words

We normalize image's $[width, height]$ to $[1,000, 1,000]$ pixels and use off-the-shelf OCR engine[2] for word recognition. Each word $w_i$ is identified by text $s_i$ and bounding box $b_i = (x_{min}, y_{min}, x_{max}, y_{max})_i$. $(x_{min}, y_{min})_i$ and $(x_{max}, y_{max})_i$ are coordinates of the top-left and bottom-right corners of $i$-th word's bounding box. The recognized words are arranged "left-to-right, top-to-bottom" and are tokenized to form the input token sequence $T = \{t_i | i \in [1, L]\}$, where $L$ is the total number with a max length limit of 512.

[2]https://learn.microsoft.com/en-us/azure/applied-ai-services/form-recognizer/?view=form-recog-3.0.0

## 2.2 Layout-Aware Encoder

Similar to Xu et al. (2021), the $i$-th token $t_i$'s embedding $x_i \in \mathbb{R}^H$ consists of token embedding $x_i^{text}$, 1D position embedding $x_i^{pos}$, and 2D layout embedding $x_i^{layout}$:

$$x_i = \text{LayerNorm}(x_i^{text} + x_i^{pos} + x_i^{layout}) \quad (1)$$

**Text Embedding & 1D Position Embedding.** The look-up table and absolute position are used for each token's text and 1D position embedding:

$$x_i^{text} = \text{EmbText}(t_i); \quad x_i^{pos} = \text{Emb1D}(i) \quad (2)$$

**2D Layout Embedding.** We introduce a 2D layout representation $x_i^{layout} \in \mathbb{R}^H$ that encodes $i$-th

token's visual layout:

$$x_i^{layout} = \text{Linear}([(\text{Emb}_x(x_{min}, x_{max}, w)_i; \\ \text{Emb}_y(y_{min}, y_{max}, h)_i)]) \quad (3)$$

where $[\cdot; \cdot]$ is the concatenation operator, $\text{Linear}(\cdot)$ is a linear projection layer to map embedding dimension to $H$. $w$ and $h$ are word bounding box's width and height. For a word tokenized into multiple sub-word tokens, we assign its bounding box to each token as an approximation due to OCR's inability for sub-word recognition.

The layout-aware encoder (LAE) is based on Transformer (Zhao et al., 2023; Vaswani et al., 2017) encoder architecture and produces layout-aware representations $x_i^* = \text{LAE}(x_i)$ for each token.

## 2.3 Multi-Step Conductive Decoder

Three step-decoders are unified through the conduction of hidden states to form the multi-step conductive decoder and fulfill the decoding chain in a three-step end-to-end manner.

### 2.3.1 Reordering Step-Decoder

With layout-aware representations $x^*$ as context memory and previously generated reading order indexes, the reordering step-decoder (RSD) autoregressively produces hidden states $h_i^{rsd}$ and reading order index $\hat{Idx}_i$ of current input token $t_i$:

$$h_i^{rsd} = \text{RSD}(x^*, t_{<i}) \quad (4)$$

$$P_i^{rsd} = \text{Softmax}(\text{Linear}(h_i^{rsd})) \quad (5)$$

$$\hat{Idx}_i = \text{Argmax}(P_i^{rsd}) \quad (6)$$

where $P_i^{rsd}$ is the classification probability for $t_i$'s reading order index over $L$ (input length) index classes, meaning that the $\hat{Idx}_i$-th token of input sequence should be "read" at $i$-th decoding step.

The reordering step-decoder is based on Transformer (Zhao et al., 2023; Vaswani et al., 2017) decoder architecture. Similar to the layout-aware encoder, it encodes tokens' bounding boxes at the embedding layer to make use of the 2D layout feature.

### 2.3.2 Segmentation Step-Decoder

The hidden states $h^{rsd}$ outputted by RSD can be viewed as the semantic representations of an in-order token sequence. Therefore, they are utilized to segment sentences from the decoded token sequence in a sequence-labeling manner. Specifically,

LayoutDIT's segmentation step-decoder (SSD) employs a lightweight Transformer (Zhao et al., 2023; Vaswani et al., 2017) encoder for context encoding and a linear projection for token-level tagging:

$$h_i^{ssd} = \text{SSD}(h_i^{rsd}) \quad (7)$$

$$P_i^{ssd} = \text{Softmax}(\text{Linear}(h_i^{ssd})) \quad (8)$$

$$\hat{B}_i = \text{Argmax}(P_i^{ssd}) \quad (9)$$

where $P_i^{ssd}$ is $i$-th token's classification probability over pre-defined 2 sentence boundary classes - {*BOS*, *IOS*}, which represents the *beginning* and *inside* of a sentence, respectively.

Based on the decoded sentence boundary tags $\{\hat{B}_i\}_{i=1}^L$, the representation sequence $\{h_i^{rsd}\}_{i=1}^L$ is segmented into $M$ sub-sequences $\{h_k^{sent}\}_{k=1}^M$:

$$h_k^{sent} = \mathcal{R}(\{h_i^{rsd}\}_{i=1}^L, \{\hat{B}_i\}_{i=1}^L, k) \quad (10)$$

where $\mathcal{R}(\cdot)$ is the sentence segmentation rule that searches for $BOS_k$ and $BOS_{k+1}$ within $\{\hat{B}_i\}_{i=1}^L$ and extracts sub-sequence $h_k^{sent}$ from $\{h_i^{rsd}\}_{i=1}^L$ between $[BOS_k, BOS_{k+1})$.

### 2.3.3 Translation Step-Decoder

Taking $h_k^{sent}$ as the context memory of the $k$-th source sentence, the translation step-decoder (TSD) generates each sentence's translation autoregressively. Specifically, at the $j$-th decoding step of $k$-th source sentence, we have:

$$h_{k,j}^{tsd} = \text{TSD}(h_k^{sent}, y_{<j}) \quad (11)$$

$$P_{k,j}^{tsd} = \text{softmax}(\text{Linear}(h_{k,j}^{tsd})) \quad (12)$$

$$\hat{Y}_{k,j} = \text{BeamSearch}(P_{k,j}^{tsd}, P_{k,<j}^{tsd}, \hat{Y}_{k,<j}) \quad (13)$$

TSD is based on the Transformer (Zhao et al., 2023; Vaswani et al., 2017) decoder architecture. Translations of all source sentences are generated in parallel to promote training and inference efficiency.

The final document translation can be obtained by concatenating all sentences' translations.

## 2.4 Optimization

LayoutDIT is optimized with supervision from all three decoding steps as follows:

$$\mathcal{L}_{rsd} = \sum_{i=1}^L \text{CE}(Idx_i, P_i^{rsd})/L \quad (14)$$

$$\mathcal{L}_{ssd} = \sum_{i=1}^L \text{FocalLoss}(B_i, P_i^{ssd})/L \quad (15)$$

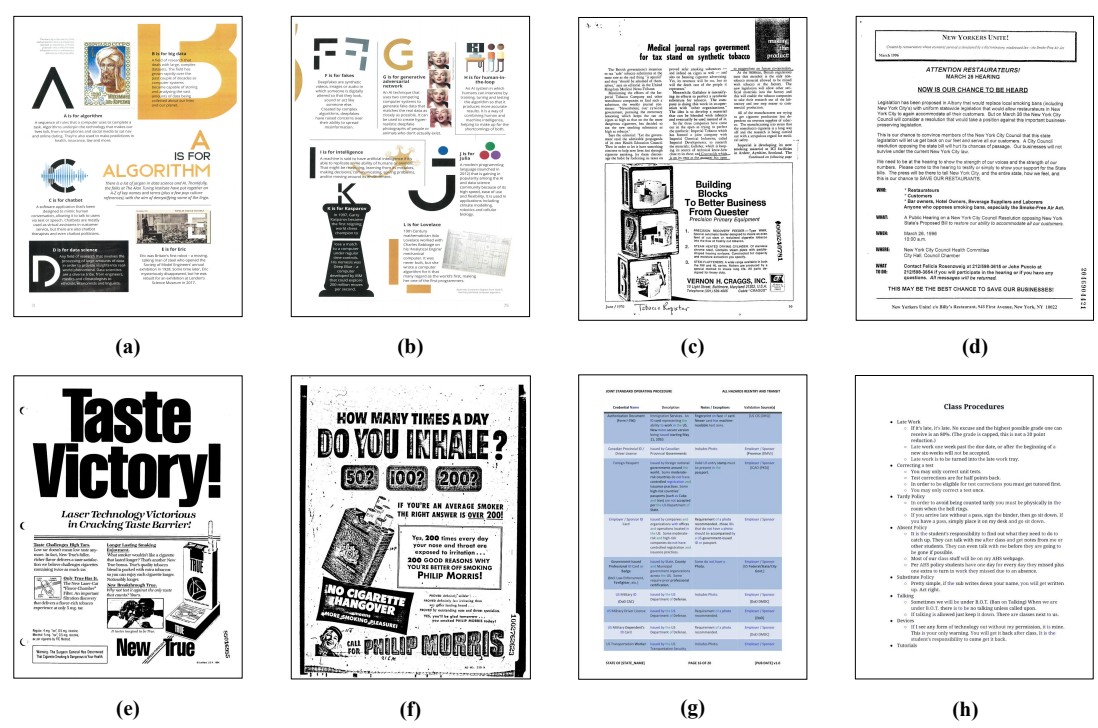

Figure 4: **Dataset samples.** (a) (b) (c) (d) (e) (f): From DITrans. (a) (b): *Political report*. (c) (d): *Newspaper*. (e) (f): *Advertisement*. (g) (h): From ReadingBank.

| Dataset | ReadingBank | DITrans | | |
|---|---|---|---|---|
| Domain | General | Political Report | Newspaper | Advertisement |
| # Page | 410K | 902 | 396 | 377 |
| # Word Avg. | 196 | 245 | 219 | 123 |
| # Sent. Avg. | 21 | 24 | 16 | 16 |
| Train/Test (# Page) | 400K/10K | 790/112 | 347/49 | 330/47 |
| Train/Test (# Sent.) | 8,467K/211K | 19,210/2,698 | 5,501/929 | 5,303/815 |

Table 1: **Datasets Statistics.** # Word/Sent. Avg. means average # word/sentence of each document image.

$$\mathcal{L}_{tsd} = \sum_{k=1}^{M} \sum_{j=1}^{|Y_k|} \text{CE}(Y_{k,j}, P_{k,j}^{tsd}) / \sum_{k=1}^{M} \sum_{j=1}^{|Y_k|} j \quad (16)$$

Here $Idx_i$, $B_i$, $Y_{k,j}$ refer to the ground-truth labels of the reading order index, sentence boundary class, and target-language token, respectively. CE($\cdot$) refers to CrossEntropy. FocalLoss($\cdot$) (Lin et al., 2017) is introduced to alleviate the class-imbalance issue. Sum of these loss functions forms LayoutDIT's total training loss:

$$\mathcal{L}_{tol} = \mathcal{L}_{rsd} + \mathcal{L}_{ssd} + \mathcal{L}_{tsd} \quad (17)$$

## 3 Multi-Domain DIT Dataset

To the best of our knowledge, there is only one publicly available dataset named ReadingBank (Wang et al., 2021) that consists of web-crawled general-domain document images with pseudo labels. Therefore, we first manually annotate a multi-domain DIT dataset named **DITrans** to test Layout-DIT's domain-specific capability. DITrans contains sophisticatedly selected images of three domains: *political report*, *newspaper*, and *advertisement*. We hire 35 professional annotators to label the reading order and sentence boundaries and 30 translators for document translation. Translators are shown a document image with reading order and sentence boundaries and are required to produce correct and fluent translations for them in Chinese. In addition, We hire 8 professional annotators to sample and check the annotated instances for quality control. We annotate 1,675 instances in total with 34,456 parallel sentence pairs. Figure 4 presents document image examples from our DITrans and the public ReadingBank. Table 1 gives a cursory review of the two datasets.

## 4 Experiments

### 4.1 Datasets and Evaluation Protocal

Experiments are conducted on ReadingBank and our DITrans for the en-zh DIT task. The original split of ReadingBank is `Train: Test = 400K:50K`. As shown in Table 1, we sample 10K from its 50K test examples as the testset. For DITrans, each

domain is split with the ratio `Train: Test = 7:1`.

We evaluate model performance with average page-level BLEU (Papineni et al., 2002) that refers to the micro-average precision of n-gram overlaps within a page between model prediction and ground-truth reference.

## 4.2 Settings

We use Transformer's encoder/decoder layer to build LayoutDIT's components. Concretely, LayoutDIT's LAE/RSD/SSD/TSD consists of 6/6/1/6 layers, respectively. Each layer has 768-dimensional hidden sizes, 12 attention heads, and 3,072 feed-forward hidden units.

We continue training LayoutDIT on DITrans' experiments after pretraining it on the large-scale ReadingBank. Adam optimizer (Kingma and Ba, 2015) is applied for training with $\beta_1 = 0.9$, $\beta_2 = 0.98$. The learning rate is $7 \times 10^{-5}$ with a linear schedule strategy. Both the dropout rate and label smoothing value are set to 0.1. The training of LayoutDIT lasts for 5 epochs with a batch size of 30 for all training runs.

## 4.3 Baselines

We consider existing cascade methods as baselines:

- **DocHandler** (Sable et al., 2023). **DocHandler-1**: It enhances DocHandler's OCR-Translation cascade by inserting a rule-based reading order detector ("left-to-right, top-to-bottom") & an unsupervised sentence segmenter (Kiss and Strunk, 2006) between them. **DocHandler-2**: It replaces DocHandler-1's sentence segmenter with a supervised Transformer-based sequence labeler.
- **MGTrans** (Hinami et al., 2021). It employs a layout parser for layout boxes detection and equips a rule-based reading order detector, sentence segmenter, and translator. We employ two SOTA layout parsing frameworks: 1) Convolution-based Cascade-RCNN (Cai and Vasconcelos, 2018), denoted as **MGTrans-Conv**. 2) Transformer-based DETR (Zhu et al., 2020), denoted as **MGTrans-DETR**.

We also build two strong end-to-end baselines to compare with our proposed LayoutDIT.

- **LayoutLM-Dec**. LayoutLM (Xu et al., 2020), a multi-modal document image understanding model that is pre-trained on large-scale IIT-CDIP (LEWIS et al., 2006) document images,

| # | Method | End-to-End | # Params (M) | BLEU |
|---|--------|------------|--------------|------|
| 1 | DocHandler-1 (Sable et al., 2023) | - | 142 | 30.47 |
| 2 | DocHandler-2 (Sable et al., 2023) | - | 172 | 37.75 |
| 3 | LayoutLM-Dec | ✓ | 232 | 45.54 |
| 4 | LiLT-Dec | ✓ | 250 | 45.79 |
| 5 | LayoutDIT-Cascade | - | 293 | **48.29** |
| 6 | LayoutDIT | ✓ | 206 | **48.20** |

Table 2: **Results on ReadingBank.** *LayoutDIT* significantly outperforms exiting cascades and end-to-end baselines. It also achieves competitive results with a lighter model architecture compared with its cascade variant LayoutDIT-Cascade.

is employed as the encoder and is equipped with a text decoder for translation.
- **LiLT-Dec**. It replaces LayoutLM with the dual-stream LiLT (Wang et al., 2022) encoder that shows stronger document image understanding ability.

For fair comparisons, the numbers of encoder/decoder layers for LayoutLM-Dec and LiLT-Dec are 12 and share the same # hidden size, # attention head and # feed-forward unit as that of LayoutDIT. All the training hyperparameters (batch size, training epochs, learning rate, etc) are kept the same as LayoutDIT. We also continue training all baselines on DITrans' experiments after pretraining them on ReadingBank.

## 4.4 Main Results

**On ReadingBank**. The results are shown in Table 2. Since ReadingBank has no layout annotations for training a layout parser, we exclude MGTrans in this experiment. LayoutDIT-Cascade is LayoutDIT's cascade variant that explicitly outputs all intermediate results (tokens or tags) during decoding sub-processes. It also replaces LayoutDIT's translation decoder with an encoder-decoder to encode tokens, thus increasing model parameters. Our proposed LayoutDIT significantly outperforms existing methods by improving BLEU from 37.75 to 48.20. Compared with LayoutDIT-Cascade, LayoutDIT achieves competitive results with a lightweight model size (87M fewer parameters), demonstrating its parameter efficiency because training and maintaining a cascade is costly.

**On DITrans**. The following two experimental settings are conducted on DITrans to verify LayoutDIT's effectiveness.

| # | Method | End-to-End | # Params (M) | Multi-Domain Learning | | | Zero-Shot Cross-Domain Transfer | | |
|---|--------|-----------|--------------|------------------|-----------|---------------|-------------------------|--------------------|------------------------|
| | | | | Political Report | Newspaper | Advertisement | Political Report (SD) | Newspaper (TD.1) | Advertisement (TD.2) |
| 1 | DocHandler-1 (Sable et al., 2023) | - | 142 | 21.47 | 26.28 | 23.14 | 21.80 | 16.87 | 13.55 |
| 2 | DocHandler-2 (Sable et al., 2023) | - | 172 | 24.80 | 27.34 | 24.99 | 25.19 | 16.88 | 13.60 |
| 3 | MGTrans-DETR (Hinami et al., 2021) | - | 212 | 28.30 | 27.61 | 25.76 | 27.34 | 16.26 | 13.09 |
| 4 | MGTrans-Conv (Hinami et al., 2021) | - | 238 | 28.99 | 28.84 | 25.63 | 29.51 | 17.78 | 14.07 |
| 5 | LayoutLM-Dec | ✓ | 232 | 45.86 | **40.66** | 36.90 | 45.71 | 27.56 | 21.40 |
| 6 | LiLT-Dec | ✓ | 250 | 46.11 | 40.37 | 36.22 | 46.61 | 23.71 | 18.14 |
| 7 | LayoutDIT-Cascade | - | 293 | 36.78 | 34.56 | 35.36 | 39.85 | 21.89 | 16.62 |
| 8 | LayoutDIT | ✓ | 206 | **46.97** | 38.82 | **45.72** | **46.70** | **29.01** | **29.46** |

Table 3: **Results on DITrans.** SD/TD means source/target domain. For DITrans' three domains, **LayoutDIT** shows the best results compared with existing cascade methods. Despite a lightweight model size, it also achieves better or competitive results compared with end-to-end baselines, in both multi-domain learning and cross-domain transfer.

| # | Model | BLEU |
|---|-------|------|
| 1 | LayoutDIT | **48.20** |
| 2a | w/o 2D layout Embedding | 43.87 |
| 2b | w/o 1D position Embedding | 39.40 |
| 3a | w/o reading order decoding loss | 3.15 |
| 3b | w/o decoding processes decomposition | 13.99 |
| 4 | w/ SSD's hidden states as context memory | 45.85 |

Table 4: Ablation study of our model on ReadingBank.

**1) Multi-Domain Learning**. In this setting, models are trained on DITrans' all three domains and tested on each domain. As shown in Table 3, our proposed LayoutDIT achieves the best results, outperforming previous cascade methods by a substantial margin. Moreover, It also shows better or comparable results than end-to-end baselines with a lighter architecture, demonstrating its excellent capability in domain-specific DIT conditions.

**2) Zero-Shot Cross-Domain Transfer.** In this setting, models are trained only on *political report* and are tested on each domain. As shown in Table 3, LayoutDIT achieves state-of-the-art results for all three domains, demonstrating that it can better transfer knowledge from the seen domain to the unseen domain in zero-shot DIT scenarios.

## 4.5 Ablation Study

To investigate the effectiveness of different components, we further compare LayoutDIT with several variants in Table 4.

**On the Input Embedding.** Removing LayoutDIT's 2D layout embedding or 1D position embedding results in model #2a/#2b. Comparing #1, #2a, #2b shows that: 1) 2D layout is useful for LayoutDIT to understand document layouts for translation improvement from model #2a's 43.87 to model #1's 48.20. 2) Incorporating 2D layouts and text semantics w/o. 1D position can give an accept-

able translation performance (39.40) as model #2b shows, which may be because the 2D layout can guide LayoutDIT to generate a reasonably good reading order. However, the performance drop from model #1's 48.20 to model #2b's 39.40 indicates the requirement of 1D position for the translation step-decoder. Combining the three features produces the best results.

**On the Multi-Step Conductive Decoding and Step-Decoder Supervision.** Model #3a abandons the supervision from the reordering step-decoder during training. Model #3b disables the multi-step conductive decoding paradigm and utilizes a Vallina encoder-decoder to directly transform the out-of-order source word sequence to document translation. Compared with model #1, the drastic performance drops of model #3a and #3b prove the effectiveness of LayoutDIT's multi-step conductive decoding and multi-task learning strategy.

**On the Translation Context Memory.** Layout-DIT's translation context memory is from reordering step-decoder (RSD) hidden states instead of segmentation step-decoder (SSD). Our conjecture for this is that RSD's hidden states are for token index prediction while SSD's are for sentence boundary classification. So the former is closer to source text understanding and should act as the translation context memory. This assumption is empirically confirmed by comparing model #1 and #4.

## 4.6 Low-Resource DIT

Benefiting from the joint optimization of components, end-to-end models have always shown superiority (Bansal et al., 2018; Tu et al., 2019) than cascades in low-resource conditions, which is particularly challenging for DIT due to the expensive annotation of document images. Therefore, we compare LayoutDIT's few-shot learning ability on DITrans *political report* with the cascade method

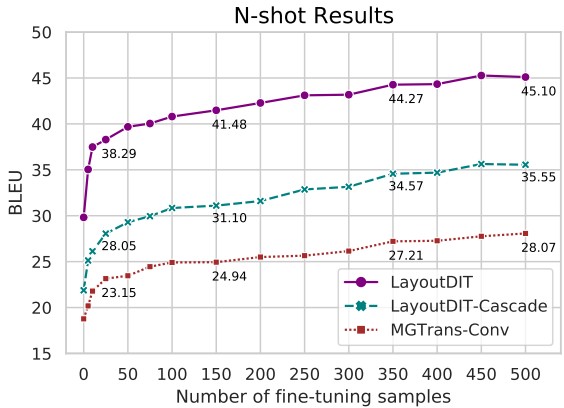

Figure 5: Comparison of few-shot learning.

| Method | ReadingBank En-De | ReadingBank En-Fr |
|---|---|---|
| DocHandler-2 | 17.15 | 18.23 |
| LiLT-Dec | 30.60 | 36.00 |
| LayoutDIT | 33.25 | 40.09 |

Table 5: Results on En-De and En-Fr DIT tasks. Evaluation metric is BLEU.

MGTrans-Conv.

As shown in Figure 5, LayoutDIT-Cascade constantly outperforms MGTrans by a large margin. Notably, LayoutDIT-Cascade achieves almost equivalent performance (28.05) using 25 samples than MGTrans does (28.07) using 500 samples. LayoutDIT brings further improvement on top of LayoutDIT-Cascade, showing great superiority in low-resource DIT conditions.

### 4.7 Evaluation on More Languages

We further evaluate LayoutDIT on English-German (En-De) and English-French (En-Fr) document image translation. We first use document images downloaded from ReadingBank (Wang et al., 2021) to construct the synthetic translation labels with Google Translate API, resulting in the En-De and En-Fr versions of ReadingBank. All models are trained for 80K steps with a batch size of 10. Table 5 shows the results of our proposed LayoutDIT on these two language pairs. Again we observe that our model substantially outperforms the cascaded DocHandler-2 by 16.10/21.86 BLEU and the end-to-end LiLT-Dec by 2.65/4.09 BLEU on En-De and En-Fr directions, respectively.

### 4.8 Case Analysis

To further reveal LayoutDIT's advantages, we provide a translation example of LayoutDIT and MGTrans-Conv in Figure 6. As shown in (a), MG-

Trans misrecognizes several small layout boxes to be a large layout (red box). This falsely parsed layout disrupts the overall box-level reading order. Moreover, as shown in (b), because of MG-Trans' "left-to-right, top-to-bottom" reading order within a layout box, some words are placed in the wrong position, causing further errors propagated to sentence segmenter. The accumulated errors ultimately cause mis-segmented, semantically-confusing source sentences and a failed translation, as shown in (c). On the contrary, by incorporating layout information, our LayoutDIT correctly captures the reading logic of words. The subsequent sentence boundary decoding and translation decoding also exhibit better results, validating the effectiveness of layout incorporation in LayoutDIT's layout-aware encoder and joint optimization of its step-decoders for minimal error accumulation.

## 5 Related Work

**Sentence Image Translation.** Sentence Image Translation (SIT) aims to translate an image with a single embedded sentence from one language to another. It is typically achieved by joining the OCR and translator. Recently, end-to-end models (Ma et al., 2023a; Ma et al., 2023b; Ma et al., 2023c; Ma et al., 2022; Chen et al., 2022; Chen et al., 2021) have been proposed to address cascades' error propagation and parameter redundancy. One approach is taking the recognition model for SIT like TRBA (Baek et al., 2019). To alleviate the end-to-end data scarcity, multi-task learning (Ma et al., 2022; Chen et al., 2021; Su et al., 2021) is leveraged to incorporate external OCR/MT datasets for training end-to-end models. MHCMM (Chen et al., 2022) further enhances feature representation through cross-modal mimic learning with additional MT data.

Although end-to-end SIT models have shown acceptable performance compared with their cascade counterparts, all of them presuppose that the sentence bounding boxes of a document image are provided by annotation and only focus on the cropped sentence images translation, leaving the more practical document image translation under-explored.

**Document Image Translation.** Existing DIT methods (Sable et al., 2023; Hinami et al., 2021; Shekar et al., 2021; Afli and Way, 2016) are scarce and are almost cascade methods, suffering layout-unaware and error propagation problems. The first attempt (Jain et al., 2021) at constructing an image-

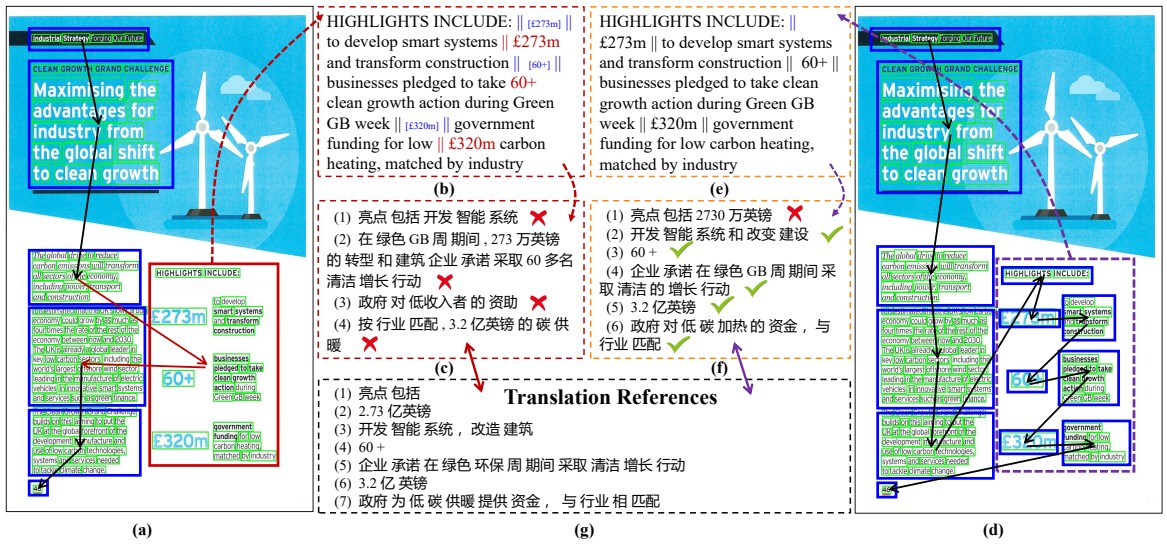

Figure 6: **A *political report* translation example of MGTrans-Conv and LayoutDIT.** (a): Parsed layout boxes and reading order produced by MGTrans. Red color denotes MGTrans' errors. (b): Segmented sentences from the red box in (a). Red color denotes MGTrans' errors of word order and sentence boundaries "‖". Blue color means ground-truth labels. (c): MGTrans' translation results. (d): Reading order decoded by LayoutDIT. For clarity, we post-process LayoutDIT's word-level reading order to box-level. (e): Sentence boundaries decoded by LayoutDIT. It only misses one sentence boundary denoted in blue. (f): LayoutDIT's translation results. (g): Translation references.

to-translation end-to-end DIT model is based on a Vanilla visual-encoder-decoder framework and only focuses on one-column regular-layout document images. It requires massive data for training and fails to handle real-world complex-layout document images.

Different from them, LayoutDIT goes beyond and pursues the end-to-end translation of arbitrary-layout document images, effectively and efficiently.

## 6 Conclusion

DIT has been shown unsatisfactory performance due to the struggling layout incorporation and cascade sub-module isolation of existing methods. In this work, we propose LayoutDIT to alleviate these two issues. It utilizes a layout-aware encoder for layout-text joint understanding and a multi-step conductive decoder for step-by-step translation decoding. Benefiting from the layout-aware end-to-end modeling, LayoutDIT significantly surpasses previous approaches and promotes DIT to a higher performance level. In the future, we will explore incorporating OCR and images' vision features into our framework to realize a more efficient and powerful DIT.

## Limitations

Since our model involves an additional step of OCR, its model size could be further compressed to achieve OCR-free DIT with the incorporation of OCR models, which will be our future exploration. Besides, our model leaves the document image's vision information to be exploited, which is an important clue for better document image understanding. We will conduct experiments with LayoutLMv2/v3 for image feature incorporation in our future works.

## Acknowledgements

We are grateful to all annotators for constructing DI-Trans. This work is supported by the National Natural Science Foundation of China (NO. 62106265).

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
