# OpenReview forum: "LayoutDIT: Layout-Aware End-to-End Document Image Translation with Multi-Step Conductive Decoder"
_EMNLP/2023/Conference — EMNLP 2023 Findings_

### Official Review · Reviewer_iJGw · 2023-07-31

**Soundness:** 3

**Excitement:**

2: Mediocre: This paper makes marginal contributions (vs non-contemporaneous work), so I would rather not see it in the conference.

**Missing References:**

- Long, Shangbang, et al. "Towards end-to-end unified scene text detection and layout analysis." Proceedings of the IEEE/CVF Conference on Computer Vision and Pattern Recognition. 2022.

**Paper Topic And Main Contributions:**

This ppaer propose a layout-aware end-to-end document translation framework named LayoutDIT, with a novel layout-aware encoder and
multi-step conductive decoder to enable layout awareness and sub-module interactions simultaneously, which could share computation among four complementary tasks, including layout parsing, reading order detection, sentence segmentation, and translation. This paper also curated a new multi-domain DIT dataset named DITrans.

**Reasons To Accept:**

It's an end-to-end document image translation model, probably first of its kind.

**Reasons To Reject:**

Lack of novelty:
(1) The novelty of the so-called layout-aware encoder lies in the combination of 1D, 2D, pos embedding and word embeddding. However, this is just a standard practice in most document layout analysis tasks so the novelty is limited.

**Reproducibility:**

4: Could mostly reproduce the results, but there may be some variation because of sample variance or minor variations in their interpretation of the protocol or method.

**Reviewer Confidence:**

2: Willing to defend my evaluation, but it is fairly likely that I missed some details, didn't understand some central points, or can't be sure about the novelty of the work.

---

> ### Author Rebuttal · Authors · 2023-08-29
>
> We sincerely thank you for the valuable review and the time dedicated to reviewing our paper.
>
> #### 1. On the novelty of our method.
>  * Our novelty mainly lies in the original DIT framework, which indeed for the first time achieves end-to-end DIT for complex-layout document images.
>  * For its decoding phase, we innovatively design a multi-step conductive decoder to unify reading order detection, sentence segmentation and ultimate translation. It contributes a lot to end-to-end DIT, comparing our LayoutDIT with LayoutLM-Dec in Table 2 & Table 3.
>  * For its encoding phase, we make the first attempt to insert the layout knowledge into DIT. The layout-aware encoder is not limited to the 2D-style layout box embedding and could be easily extended to other advanced layout-aware strategies, e.g., from document layout analysis tasks as you recommend. There are alternative implementations of the layout-aware encoder, and in this paper, we follow the simple and effective layout embedding strategy in LayoutLM [1] as an instantiation, as described in section 2.2. Following your suggestions, we will conduct experiments with more layout strategies from document layout analysis tasks.
>
> #### 2. About the supplemental reference
>  * Thanks for your recommendation about the literature [2]. We have carefully inspected its task and model which focuses on unified word detection and word clustering (layout analysis) for scene text images, while our framework pursues end-to-end translations for document images with long document text and complex layouts. In this regard, they are two orthogonal contributions to complementary tasks. We will add the missing reference in the revised version.
>
> > [1] LayoutLM: Pre-training of Text and Layout for Document Image Understanding, KDD 2020
>
> > [2] Towards End-to-End Unified Scene Text Detection and Layout Analysis, CVPR 2022

---

### Official Review · Reviewer_ZMNX · 2023-08-04

**Typos Grammar Style And Presentation Improvements:** In line 44, "Given" -> "given"
**Soundness:** 3

**Excitement:**

3: Ambivalent: It has merits (e.g., it reports state-of-the-art results, the idea is nice), but there are key weaknesses (e.g., it describes incremental work), and it can significantly benefit from another round of revision. However, I won't object to accepting it if my co-reviewers champion it.

**Paper Topic And Main Contributions:**

This paper studies document image translation. To address the error propagation problem, the authors propose LayoutDIT with a three-step conductive layout-aware decoder including reordering, segmentation and translation sub-decoders. These multi-step conductive decoders can be optimized together and outperform previous baselines of isolated components. This paper also brings a new dataset DITrans, for document image translation. Experiments show that the overall performance of LayoutDIT is promising.

**Questions For The Authors:**

1. Will you release the DITrans dataset for community reproduction?

2. Technical questions:

(1) I would like to know the decoder configurations in LayoutLM-Dec and LiLT-Dec. This can help the readers compare LayoutDIT to the baselines. It would be better to list the detailed decoder configurations in the experiment section.

(2) Have you tried LayoutLMv2-Dec as the baseline given that LayoutLM-Dec is compared? Because LayoutLMv2 [2] should be a better encoder backbone compared to LayoutLM [1].


[1] LayoutLM: Pre-training of Text and Layout for Document Image Understanding, KDD 2020

[2] LayoutLMv2: Multi-modal Pre-training for Visually-Rich Document Understanding, ACL 2021

**Reasons To Accept:**

1. This paper is well-written and easy to understand.

2. Considering layout information for document image translation is reasonable. The three-step conductive decoder can implement modularized word reordering, sentence segmentation and ultimate translation. It is a clear and rational model architecture.

3. The experiments prove that LayoutDIT can obtain SOTA and generalized performance.

**Reasons To Reject:**

As far as I know, some popular OCR tools can also achieve reordering (step 1) and segmentation (step 2) functions, and their reordering and segmentation accuracies are satisfactory. For example, ERNIE-Layout [1] uses the preprocessed span-based OCR (reordered and segmented) to perform large-scale pretraining. Directly using span-based OCR instead of sequence-based MSOCR may further alleviate the error propagation of reordering and segmentation. LayoutDIT also relies on OCR preprocessing, it seems that RSD and SSD can be replaced by off-the-shelf span-based OCR tools.


[1] ERNIE-Layout: Layout Knowledge Enhanced Pre-training for Visually-rich Document Understanding

**Reproducibility:**

3: Could reproduce the results with some difficulty. The settings of parameters are underspecified or subjectively determined; the training/evaluation data are not widely available.

**Reviewer Confidence:**

4: Quite sure. I tried to check the important points carefully. It's unlikely, though conceivable, that I missed something that should affect my ratings.

---

> ### Author Rebuttal · Authors · 2023-08-29
>
> We sincerely thank you for your valuable reviews and for pointing out typos. We will carefully check it in the revision.
> #### 1. Comparison with span-based OCR and the ERNIE-Layout.
>  * Thanks for your valuable recommendations about span-based OCR and the ERNIE-Layout literature. We have carefully inspected the ERNIE-Layout paper, especially the details about the preprocessing in its sections 3.1 and A.1. Actually, we have experimented with MGTrans, which shares nearly the same workflow with ERNIE-Layout. Both of them fall into the conventional cascade: OCR for word recognition -> layout parser for layout boxes detection -> rule-based layout block reordering, except that MGTrans is equipped with an additional sentence segmenter since it is for the translation task. In this regard, the case in Figure 6 straightforwardly displays that MGTrans’ cascade convention suffers error accumulation at every step. Experimental results from Table 3 and Figure 5 also give a statistical verification of MGTrans’ inferior performance compared with our end-to-end LayoutDIT.
> * As for the span-based OCR, we further conduct a non-exhausted examination of some mainstream OCR engines including the commercial [MSOCR](https://azure.microsoft.com/en-ca/products/form-recognizer), [BaiduOCR](https://ai.baidu.com/tech/ocr/general), [TencentOCR](https://cloud.tencent.com/product/ocr), and the open-source [EasyOCR](https://github.com/JaidedAI/EasyOCR) and [TesseractOCR](https://github.com/tesseract-ocr/tesseract). Most of them only return a batch of recognized words or cropped text lines arranged in “left-to-right, top-to-bottom” order. This observation may explain our motivation to explicitly bring RSD and SSD into our framework as a solution for the unavailable external tools and in pursuit of sub-module joint optimization.
>  * We also want to give a justification from an empirical optimization perspective. Compared with the fixed, unlearnable external tools, our learnable RSD and SSD modules could benefit from the sub-module interaction to reduce error accumulation, especially for the adaptation to low-resource document image layouts (Figure 5).
>
> #### 2. On the DITrans dataset.
>  * Yes, we will release our DITrans to the research community through the HuggingFace platform, for reproduction of this paper. We will also keep enriching it with more domains and layout formats to facilitate long-term DIT research.
>
> #### 3. Answers to technical questions.
> ##### 3.1. On the decoder configurations.
>  * Detailed configurations of LayoutDIT have been described in section 4.2. For fair comparisons, both decoders of LayoutLm-Dec and LiLT-Dec are 12-layer Transformer decoders and share the same # hidden size, # attention head and # feed-forward units as that of LayoutDIT. We will list all parameters about model configurations and training runs in the final 9-page revision.
> ##### 3.2. On the comparison between our model and LayoutLmv2-Dec.
>  * We have experimented with the canonical LayoutLM [1] and the state-of-the-art layout model LiLT [2] which shows significant improvement compared with LayoutLMv2 [3]. Following your suggestions, we will add experiments about LayoutLMv2 as well. Thanks for your valuable advice.
>
> > [1] LayoutLM: Pre-training of Text and Layout for Document Image Understanding, KDD 2020
>
> > [2] Lilt: A Simple yet Effective Language-Independent Layout Transformer for Structured Document Understanding, ACL 2022
>
> > [3] LayoutLMv2: Multi-modal Pre-training for Visually-Rich Document Understanding, ACL 2021

---

### Official Review · Reviewer_sqAx · 2023-08-05

**Soundness:** 2

**Excitement:**

3: Ambivalent: It has merits (e.g., it reports state-of-the-art results, the idea is nice), but there are key weaknesses (e.g., it describes incremental work), and it can significantly benefit from another round of revision. However, I won't object to accepting it if my co-reviewers champion it.

**Paper Topic And Main Contributions:**

The paper presents a novel end-to-end document image translation (DIT) framework called LayoutDIT, which aims to improve the accuracy of translating text within document images from one language to another. The authors make several key contributions that address limitations in existing methods:

1. They introduce the first end-to-end DIT framework that incorporates visual layout information, avoiding error propagation seen in traditional cascade approaches.

2. To jointly model text semantics and 2D layout features, they propose a layout-aware encoder, which enhances the overall translation process.

3. The authors design a multi-step conductive decoder with three step-decoders to handle reading order, sentence boundaries, and translations in a unified end-to-end manner.

4. The proposed LayoutDIT achieves impressive results on the ReadingBank dataset and outperforms baselines on their newly created multi-domain DIT dataset, DITrans. It also demonstrates advantages in low-resource DIT scenarios.

**Reasons To Accept:**

The paper presents an end-to-end document image translation (DIT) framework called LayoutDIT, which aims to improve the accuracy of translating text within document images from one language to another. The authors make several key contributions that address limitations in existing methods:
1. They introduce the first end-to-end DIT framework that incorporates visual layout information, avoiding error propagation seen in traditional cascade approaches.

2. The authors design a multi-step conductive decoder with three step-decoders to handle reading order, sentence boundaries, and translations in a unified end-to-end manner.

3. The proposed LayoutDIT achieves impressive results on the ReadingBank dataset and outperforms baselines on their newly created multi-domain DIT dataset, DITrans.

**Reasons To Reject:**

1. The proposed DITrans dataset is relatively small (1675) compared to ReadingBank (400k). More analysis on the diversity and complexity of DITrans could be useful. When analyzing a dataset, the intermediate English annotations between the source documents and the translated documents are important for the quality of this dataset. But this analysis is not included.
2. The paper could benefit from more ablation studies analyzing the impact of each component (Reading order decoding, Sentence boundary decoding, and Translation decoding). For example, using existing machine translation benchmarks to provide translation performance in terms of correct and incorrect reading orders would be an insightful benchmark for comparing and analyzing translation decoding steps. Based on current experiments, it is still unclear which component has the most impact on the final translation performance.
3. Leveraging multilingual models like LayoutXLM could potentially improve generalization across languages. The current baseline uses an English-only LayoutLM model.

**Reproducibility:**

3: Could reproduce the results with some difficulty. The settings of parameters are underspecified or subjectively determined; the training/evaluation data are not widely available.

**Reviewer Confidence:**

4: Quite sure. I tried to check the important points carefully. It's unlikely, though conceivable, that I missed something that should affect my ratings.

---

> ### Author Rebuttal · Authors · 2023-08-29
>
> We sincerely thank you for the acknowledgment of our novelty and impressive results, where we propose the first end-to-end DIT framework that incorporates visual information.
> #### 1. DITrans Dataset Analysis.
>   * We appreciate your attention to the size and analysis of our DITrans dataset. Actually, our main contribution is to propose the first end-to-end DIT framework towards document images with complex layouts.
>   * DITrans and ReadingBank are used to evaluate specific-domain and general-domain performance, respectively. Compared with ReadingBank, DITrans was created to validate that the model can handle diverse and complex document images in the real world. Specifically, DITrans contains sufficient document images selected, scanned and annotated by humans with high-quality labels, while ReadingBank is created with the large-scale pseudo-labels. We will present more layout examples in the revision’s appendix.
>
> #### 2. The impact of each component.
>  * In the proposed framework, reading order decoding, sentence boundary decoding and translation decoding are indispensable in multi-step decoding. These three sub-modules divide and conquer the DIT task step by step in a serial, interrelated mode instead of a parallel, irrelevant way.
>  * As you point out, our paper could benefit from more ablation studies analyzing the impact of each component. Actually, we have already analyzed the impact of each component in Table 4. Specifically, comparing model #1 with #3a in Table 4 indicates the necessity of an optimized reordering step-decoder. Without it, the out-of-order hidden state sequence conducted to the subsequent decoders causes a severe performance drop from 48.20 to 3.15. Moreover, comparing model #1 with #3b in Table 4 verifies the necessity of reordering step-decoder and segmentation step-decoder. Without them, the translation step-decoder struggles to translate an out-of-order long document instead of in-order, short sentences, causing a performance drop from 48.20 to 13.99. Following your suggestions, based on current experiments, we will conduct more ablations to analyze the impact of each component in the revised version.
>
> #### 3. On the language generalization.
>  * Thanks for the valuable recommendation about LayoutXLM. Following your suggestion, we will evaluate the proposed framework on multilingual models, including LayoutXLM in the revision.

---

### Meta-Review · Area_Chair_Nfg2 · 2023-09-16

**Recommendation:** 3

**Metareview:**

This paper proposes an end-to-end solution to document image translation.  The encoder jointly takes text and the layout information as input.  The decoder has three step to deal with ordering, sentence boundary, and translation respectively.  Experiments show significant improvement over baselines.   The paper also present a new dataset for this research.  Reviewers raised concerns and the authors responsed in details.  I think the main contribution of the paper is clear: the end-to-end architecture / the three step decoder and the dataset.  The results are positive.

---

### Decision · Program_Chairs · 2023-10-07

**Decision:**

Accept-Findings

**Comment:**

This paper proposes an end-to-end solution to document image translation.  The encoder jointly takes text and the layout information as input.  The decoder has three step to deal with ordering, sentence boundary, and translation respectively.  Experiments show significant improvement over baselines.   The paper also present a new dataset for this research.  Reviewers raised concerns and the authors responsed in details.  I think the main contribution of the paper is clear: the end-to-end architecture / the three step decoder and the dataset.  The results are positive.